# Novel Straightening-Machine Design with Integrated Force Measurement for Straightening of High-Strength Flat Wire

**DOI:** 10.3390/s23229091

**Published:** 2023-11-10

**Authors:** Lukas Bathelt, Maximilian Scurk, Eugen Djakow, Christian Henke, Ansgar Trächtler

**Affiliations:** 1Fraunhofer IEM, Zukunftsmeile 1, 33102 Paderborn, Germany; maximilian.scurk@iem.fraunhofer.de (M.S.); eugen.djakow@iem.fraunhofer.de (E.D.); christian.henke@iem.fraunhofer.de (C.H.); 2Heinz Nixdorf Institute, Paderborn University, Fürstenallee 11, 33102 Paderborn, Germany; ansgar.traechtler@uni-paderborn.de

**Keywords:** straightening-machine design, straightening-force measurement, high-strength flat wire, punch-bending process

## Abstract

In a punch-bending machine, wire products are manufactured for a wide range of industrial sectors, such as the electronics industry. The raw material for this process is flat wire made of high-strength steel. During the manufacturing process of the flat wire, residual stresses and plastic deformations are induced into the wire. These residual stresses and deformations fluctuate over the length of the semi-finished product and have a negative effect on the final product quality. Straightening machines are used to reduce this influence to a minimum. So far, the adjustment of a straightening machine has been performed manually, which is a lengthy and complex task even for an experienced worker. This inevitably leads to the use of inefficient straightening strategies and causes high rejection rates in the entire production process. Due to a lack of sensor information from the straightening operation, application of modern feedback control methods has not been practicable. This paper presents a novel design for a straightening machine with an integrated, precise straightening force measurement. By simultaneously monitoring the position of the straightening rollers, state variables of the straightening operation can be derived. Additionally, a tension control for feeding the flat wire is introduced. This is implemented to mitigate the disturbing effects caused by irregularities in the wire-feeding process. In the results of this article, the high precision of the developed force measurement design and its possible applications are shown.

## 1. Introduction

Flat wire products undergo a series of different shape changes during production. The cold or hot rolling processes that are involved in the primary shaping and the subsequent winding onto coils for transport and storage introduce plastic deformations that cause residual stresses in the wire material. Their extent varies depending on the production processes and the environmental conditions involved [1]. Straightening machines are used to minimize the negative impact these residual deformations and stresses can have on downstream processes and therefore the final product [2,3]. To enable an effective mitigation strategy, small changes in the wire’s material properties need to be detected, evaluated, and acted upon. This requires the straightening machine to be equipped with a means to precisely measure process parameters. However, the amount of process information offered by conventional roller straighteners is strongly limited, with a broad range of industry-use apparatuses offering no integrated way of measuring roller forces and positions as they are purely mechanical systems consisting of a fixed and movable mounting block. The positioning of the rollers in these apparatuses is achieved by either moving the rollers as a unit or enabling them to be moved individually. In the former case, the rollers are mounted to a common mounting block that enables equidistant or increasing/decreasing positioning by changing the distance or angle between the fixed and movable mounting blocks. The latter case expands upon this by mounting the rollers on linear bearing blocks that can be moved individually. Locking of the desired position is typically achieved by utilizing toggle-lock or screw-driven systems [4,5]. To automate the straightening process, efforts have been made to improve upon these simple mechanical apparatuses by integrating sensors and actuators in the form of digital position displays and/or stepper motors, respectively. However, even with advanced designs, the acting roller forces are calculated rather than measured directly [6,7]. Examples of a conventional wire straightener and a semi-automated solution are shown in Figure 1.

### 1.1. Motivation

Development of the novel straightening-machine design was mandated by the desire to develop an innovative assistance system for the setup process of straightening machines as described in [8,9,10]. Conventional setup processes involve extensive trial-and-error processes or intensive material testing. These are time-consuming and expensive. Also, a lot of waste is produced during these procedures. Furthermore, in conventional straightening processes, the material is usually pre-formed more than necessary since it is not possible to react to changes in the material properties.

To provide insight into the straightening process, generate reliable data, and calculate the correct adjustment of the roller positions, the ability to detect small changes in the wire material’s properties as it is pulled off the coil is needed. The precise measurement of forces acting on the straightening rollers represents an appropriate means to achieve this. For this reason, development of an intelligent, self-optimizing straightening machine with an integrated force-measurement concept was incentivized and carried out in earlier work [10,11]. However, several points of improvement to mitigate inaccuracies within the force-measurement process during experimental use could be identified. Improving or eliminating these served as the central motivation for the development of the novel straightening-machine design including precise force measurement described in this article.

### 1.2. Review: State of the Research Field and Measuring Concepts

To enable further automation of the straightening process, various experimental research machines have been developed following different objectives and approaches. Among these are roller straightening systems, which incorporate concepts to measure the acting process forces. During the development of the straightening machine described in this article, a select batch of these research machines from related work were evaluated. Special attention was paid to the utilized force-measurement concepts, as these were paramount for the design process.

In [1], a simulation program for calculating the optimal roller settings based on the acting process forces and a corresponding control system were developed. To verify the results and collect data, a sheet-metal straightening machine capable of measuring the roller forces was used. In Figure 2, the integration method is shown in a schematic view of the upper machine housing. Movement of the upper rollers was enabled by individually raising or lowering them via a motor driven set of spindles. Measurement of the roller forces was achieved by integrating purpose-built ring-sensors at the bearing points of the drive spindles. The process force’s point of attack being in the center in between the spindle bearing points resulted in corresponding bending torques. These, in turn, led to uneven loads on the force-measurement rings due to the resulting deformations and necessitated further modifications of the measurement concept to mitigate this effect.

A very similar machine was used in [2,3,13]. However, the force measurement concept made use of conventional load cells placed in between the roller and the driving end of the spindles. This way, a direct contact point between the spindle bearings and force sensors was avoided, thus circumventing the problem of uneven load (cf. Figure 2), and reducing the complexity of the measurement concept at the same time.

A measurement concept utilizing load cells was also used by [14] during the development of a closed-loop control for a wire-bending machine. The load cells were used to acquire the roller process forces, which were then fed to the control system. To achieve this, they were integrated in the straightening machine located before the bending machine. The straightening rollers were integrated into forked bodies, which were inserted laterally into a groove plate for guidance. In addition to accommodating the straightening roller, the forked body also served as a guide for vertical movement. Above the forked body, the force-measurement sensors were integrated along with a guiding element. To adjust the position, the upper guiding elements were connected to the housing above them using tension springs, and a wedge block resting on guidance rollers was employed. Vertical movement of the rollers was then achieved by horizontally moving the wedge blocks. The linear movement required for this was created by connecting the wedge blocks to spindles driven by stepper motors [14]. Figure 3 illustrates this concept in a schematic way. In contrast to the concept of integrating the force measurement into the upper roller assembly, the work of [15,16] showed that integration into the lower roller assembly can also be feasible.

Another approach was used for a straightening machine in [10,11]. It was developed for a precursor project to the novel machine described in this article [8]. Its structure is shown in Figure 4. The straightening machine consisted of a centrally mounted seven-roller straightening machine fixed to a lower mounting plate. Vertical movement of the three upper rollers was enabled by mounting them to the lower part of the straightening units, each consisting of two linear bearing blocks with a load cell screwed in between. To provide translational movement, the upper bearing blocks were outfitted with spindle threads which enabled moving the straightening units via three spindles connected to driveshafts driven by stepper motors [11]. The machine performed well; however, during its operation, several points of possible improvement emerged.

Vertical positioning of the rollers was controlled by utilizing the stepper motor encoder signal and relating it to the thread pitch of the spindles. As a consequence of this, the roller position had to be re-calibrated frequently, as the positional information would be lost every time the machine was turned off. The design placed the mounting point of the straightening-unit rollers off-axis from the integrated load cells, leading to torques at the load cell mounting points which had a negative effect on measurement accuracy. Load cells were also threaded into the guiding elements, creating a rigid connection to transfer these torques from the rollers to the sensor. Furthermore, the linear bearing blocks possessed large overlapping guiding surfaces offering high potential for friction. As a result, a negative influence of moving the straightening units was observed. The measured roller forces peaked during movement and settled again after reaching the designated end position. To address the high resulting forces caused by this behavior, the implemented load cells possessed nominal forces of 2 kN and were capable of a force resolution of 20 N. This complicated reliable measurements, as it caused high noise and thus required extensive filtering of the sensor signal.

### 1.3. Aim and Structure of the Paper

The main aim of this work is the integration of a precise force and position measurement into a straightening machine. This is achieved by developing a novel straightening-machine design with an integrated measurement concept to approach the precise measurement of roller forces. Therefore, the conceptualization, design, and validation of the straightening machine were conducted.

In a further step, a model approach was developed which represents the essential forming operations in a straightening machine. Special focus is placed on the applicability of the model for the development of control algorithms for the straightening process. Within this context, one of the main requirements is a small computation time. Finally, the developed model was validated. For this purpose, the simulated straightening forces were compared with the measured straightening forces of the newly developed straightening machine.

Considering the state of the art, it is noticeable that there are still no adequate approaches for inline measurement of the wire curvature. However, this is one of the most important parameters for evaluating the straightening quality. For this reason, the straightening machine is considered in this paper together with the downstream process of punch-bending. Online measurement of the straightening forces and determination of the quality parameters (here α and θ) in the punch-bending process are used to determine the necessary adjustment of the straightening rollers.

In Section 2 of this paper, the straightening machine is classified as a mechatronic system in its field of application, and the system variables are described. This is followed by a detailed description of the intended control concept. Afterwards, a model approach describing the forming operation in a single straightening triangle is presented. Section 3 describes the design process of the new straightening machine. The results of the conceptional design and design implementation are presented in detail. Section 4 describes the experimental studies and their results. The decoupling of the measured variables and the sensitivity of the force measurement are verified. Afterwards, the measurement results of the new straightening machine are compared with the simulation results of the model approach.

## 2. Control Structure and Model Approach

This section describes the investigated system from a control engineering point of view. For this purpose, the system and the intended control structure with its system variables are presented. Afterwards, a model approach is presented, which is to be used at a later stage in the design of the feedforward control and closed-loop control.

### 2.1. System Description and Control Structure

The dynamic adjustment of the straightening rollers during the manufacturing process is desirable in order to be able to react to fluctuating material properties of the semi-finished product. Particular focus is placed on the development of a robust and efficient method that does not require time-consuming or resource-intensive preliminary tests and is suitable for industrial use. To develop such an approach, this paper considers the system shown in Figure 5.

The semi-finished product is uncoiled from a coil and fed to the straightening process via guide rollers. The straightening machine is equipped with sensors to determine the vertical roller positions ${\underline{z}}_{M}$ and the straightening forces ${\underline{F}}_{R}$ on the upper three rollers. The semi-finished product is pulled through the straightening machine by a feeding unit. This is followed by a material buffer to make the transition from the continuous straightening process to the discontinuous punch-bending process. The punch-bending process is used in this paper as an example for a downstream process. There are other possible processes like progressive tools which can be used instead. The punch-bending machine consists of a feeding unit and a punch unit. In this application, an L-shaped component is produced. The bending angle α and the straightness θ of the leg are recorded via a camera. They represent the quality parameters of the manufactured parts and serve as variables to be controlled within a closed-loop control algorithm. They belong to the state variables that describe the manufacturing process. The wire curvature and the stress states in the wire also belong to these. A detailed description of the dependences of the state variables is given in Section 2.2. The positions of the straightening rollers have the greatest influence on the straightening process. The straightening result is also influenced by the pre-curvature of the wire and its deformation history. Further influencing factors may be, for example, variations in the wire thickness or in the material structure. A PLC controls all the drives and processes the sensor data. The determining variables for the process speed are the coil speed $\omega_{C}$ upstream of the straightening machine, the feeding speed $\omega_{F}$ of the feeding unit downstream of the straightening machine, and the cam disk speed $\omega_{CS}$ of the punch-bending machine. No further control variables were considered in the punch-bending process. The influence of the process speed remains to be investigated. In particular, the tensile force applied to the wire due to feeding can have a significant influence on the straightening process.

The aim of the control approach presented here is the dynamic adjustment of the reference positions ${\underline{z}}_{R}$ for the straightening rollers. On the one hand, this should keep the quality of the straightening process reproducible at a high level. On the other hand, it should be possible to react to fluctuations in the material properties with a dynamic adjustment of the straightening rollers in the process. The fluctuations in the material properties are interpreted as disturbance variables and can be divided into two categories: deterministic and stochastic disturbance variables. In this approach, the change in pre-curvature over the coil radius is considered as a deterministic disturbance variable. Stochastic disturbances are, for example, width and thickness variations or the material composition.

Accordingly, the developed control approach is also divided into two parts. The feedforward control estimates and compensates for the change in pre-curvature with the aid of disturbance compensation. The feedback control is provided to correct the estimated roller positions in order to react to stochastic disturbances. The structure of the control approach is shown in Figure 6 and is based on the two-degrees-of-freedom structure (cf. [17]).

The vector $\underline{R}$ contains the setpoints for feedforward and feedback control:(1)$$\underline{R}={\left[\alpha_{R},\;\;\;\theta_{R}\right]}^{T}$$

Both $\alpha_{R}$ and $\theta_{R}$ are the reference values for quality parameters of the manufactured parts in the punch-bending machine. Depending on the configuration of the system, at least one variable must be specified as a reference. Based on the reference specification, the feedforward control specifies a reference trajectory ${\underline{z}}_{R,0}(t)$ including the positions for the three movable straightening rollers. This is corrected by the closed-loop control during the running process by $\Delta {\underline{z}}_{R}$, so that the final reference vector ${\underline{z}}_{R}$ is provided for the subordinate position control of the straightening rollers. These are controlled and act as manipulated variables on the flat wire in the first part of the controlled system, the forming process in the straightening machine. As a deterministic disturbance variable, the change of the pre-curvature $\Delta \kappa_{0}$ influences the straightening process. The totality of all stochastic disturbances is summarized in the vector $\underline{\zeta }$. This includes, for example, variations in width, thickness, or material composition. With a residual curvature $\kappa_{2}$, the flat wire is fed to the punch-bending process, which represents the second part of the controlled system. In this, an L-shaped component is formed, which is characterized by the quantities α and θ (cf. Figure 5). These quantities are measurable and can be fed to the control system as measured values.

The change of the pre-curvature $\Delta \kappa_{0}$ cannot be measured directly during the running process since it is overlaid by process-related tensile forces. This also applies to the residual curvature $\kappa_{2}$ behind the straightening machine. For this reason, only a measurement of the initial curvature $\kappa_{0}$ before starting the production process when inserting a new coil is possible via offline measurement. The initial curvature is fed to the disturbance model in the disturbance compensation. The disturbance compensation estimates the further variation of the curvature ${\tilde{\kappa }}_{0}$ depending on the coil radius $r_{C}$. In the feedforward control, a reference trajectory of the straightening roller positions as a function of time ${\underline{z}}_{R}(t)$ is calculated. A detailed description of the working principle of the disturbance compensation can be found in [9]. The algorithm of the feedback control is part of future work and will therefore not be considered here in detail.

In addition to the variables already mentioned in this control system, Figure 6 also shows the force vector ${\underline{F}}_{R}$. This vector ${\underline{F}}_{R}$ contains the signals of the straightening forces at the three movable straightening rollers, which are fed back to the control system. In previous straightening machines for flat wire, it was not possible to feed back the straightening forces because either no straightening-force measurement was provided, or this was too strongly influenced by friction effects. For this reason, the arrow is shown dashed, since the accuracy and reliability of a straightening-force measurement was investigated in the context of this paper. For this purpose, a novel design of a straightening machine for flat wire is presented, which provides for the integration of a precise force measurement. With the aid of this measurement signal, an important state variable in the straightening process is obtained for the optimal adjustment and online correction of the roller positions. This measured variable can be used to detect changes in the forming process of the straightening machine, since the straightening force is influenced by mechanical material properties such as the strength of the material. The exact relationships are described in more detail in the following section as part of the modeling approach. Nevertheless, the control of the straightening process requires further measured variables, such as α and θ, because the straightening force can be influenced by factors such as flaw structure in the wire.

This description of the system under test and of the control structure contains the configuration of all sensor information currently available at the test rig. From this point on, the focus in this paper is placed on one single straightening triangle, whereby only one roller position $z_{R}$ and one straightening force $F_{R}$ are considered.

### 2.2. Model Approach

Modeling the forming operation in a straightening machine is a highly complex task due to the interactions between the individual straightening triangles and the material behavior during alternating bending operations. For this reason, a simplified model is set up in this paper which describes the essential forming relationships and determines the resulting straightening forces as an important state variable. The model is to be used for control purposes in a later step. In this context, a more accurate model or even a FE model, such as the one used in [18], is too intensive to calculate and can only be inverted with immense effort. The shown model design essentially follows the model design in [19]. In [19], all assumptions and simplifications for the determination of strains, stresses, and bending moments are described in more detail. This will not be discussed here.

The mechanical system for the simplified model in this paper is shown in Figure 7. The *x* axis of the coordinate system points in the feed direction of the material. Any normal forces due to the feeding of the flat wire are to be disregarded for the time being. In contrast to [19], only one straightening triangle is considered in this model design. The lower rollers 1 and 3 are fixed in the z direction and the upper roller 2 is adjustable. This results in a bending line for the flat wire as shown in Figure 7b. In the red marked areas, the load stress exceeds the yield strength of the material, so that plastic deformation occurs in these edge areas. After the contact point of the wire with roller 2, the load is relieved until the wire leaves the straightening machine at roller 3.

To calculate the strain and stress curves, the bending line of the wire between the straightening rollers must be described. Previously, [19] selected an approach based on third-order polynomials for elastic regions of the bending line. It can be shown that a third-degree polynomial for the elastic bending line corresponds to the exact solution. The effort required for calculating the exact description of the bending line in regions with elasto-plastic deformation is much greater. In [1], the bending line in this region is simplified to a fourth-order polynomial. The computation of these bending lines from third- and fourth-order polynomials requires several iteration loops. This requires a lot of computational effort to determine the bending lines. Therefore, in this paper, the bending line is built from only two third-order polynomials. Thus, the following mathematical relationships apply to the bending lines $w_{12}(x)$ between straightening rollers 1 and 2 and $w_{23}(x)$ between rollers 2 and 3:(2)$$w_{12}\left(x\right)=a_{3}x^{3}+a_{2}x^{2}+a_{1}x+a_{0}$$
(3)$$w_{23}\left(x\right)=b_{3}x^{3}+b_{2}x^{2}+b_{1}x+b_{0}$$

The form of the bending lines is influenced by the position of the straightening roller 2. To determine the polynomial coefficients $a_{i}$ and $b_{i}$, an optimization problem is formulated to ensure that the bending lines do not cross the straightening rollers and are tangential to the rollers in their contact points. Therefore, a quality function J is built of the weighting factor $g_{n}$ and the term $\Delta m_{n}$:(4)$$J=\sum_{n=1}^{3}g_{n}\cdot \Delta m_{n}$$

The term $\Delta m_{n}$ describes the difference of the slope between the bending line at the contact point to roller *n* and the slope of the circular function for straightening roller n at this point. To calculate this term, a linear equation system is solved in each step of the optimization, which determines the polynomial coefficients $a_{i}$ and $b_{i}$ from Equations (2) and (3). The equation system has a total of eight unknown variables. Thus, eight equations are necessary to uniquely solve the equation system. For this purpose, C^2^ continuity is assumed at the contact points of the wire with the straightening rollers. From this, the necessary eight equations can be derived. The result after minimizing J contains the curves of the bending lines $w_{12}(x)$ and $w_{23}(x)$ between the straightening rollers.

In order to determine the strain in the wire from the form of the bending lines, two assumptions are made at this point. Firstly, only bending load on the wire is assumed, so that the neutral fiber does not receive any strain. Second, small deformations are assumed. Under these assumptions, the twofold derivative of the bending lines $w_{12}{}^{\prime \prime }(x)$ and $w_{23}{}^{\prime \prime }(x)$ can be set equal with the curvature $\kappa_{12}(x)$ and $\kappa_{23}(x)$ of the wire:(5)$$\begin{array}{c} w_{12}{}^{\prime \prime }(x)=\kappa_{12}(x)\\ w_{23}{}^{\prime \prime }(x)=\kappa_{23}(x) \end{array}$$

For calculation of strain and stress relations in the wire, the point of maximum curvature $\kappa_{max}$ in the wire is determined. Due to the continuous feeding of the wire, this point is usually located nearly before the apex of the middle roller [7]. In the bending-line approach selected here, it is located directly at the apex of straightening roller 2. The strain distribution over the wire thickness z can be determined from the maximum curvature $\kappa_{max}$:(6)$$\varepsilon \left(z\right)=-\left[z\cdot \kappa_{max}+\varepsilon_{v}\left(z\right)\right]$$

This considers the pre-strains $\varepsilon_{v}\left(z\right)$ that are already present in the wire. For the material behavior, a linear behavior is assumed both in the elastic and in the plastic area. Thus, the stress distribution can be determined from the strain distribution:(7)$$\sigma \left(z\right)=\left\{\begin{array}{ccc} E\cdot \varepsilon \left(z\right)+\sigma_{eig}(z) & ,\;if & \sigma_{Fd}\left(z\right)<\sigma \left(z\right)<\sigma_{Fz}\left(z\right)\\ \left(1-k_{E}\right)\cdot \sigma_{Fz}\left(z\right)+k_{E}\cdot \left(E\cdot \varepsilon \left(z\right)+\sigma_{eig}\left(z\right)\right) & ,\;if & \sigma \left(z\right)\geq \sigma_{Fz}\left(z\right)\\ \left(1-k_{E}\right)\cdot \sigma_{Fd}\left(z\right)+k_{E}\cdot \left(E\cdot \varepsilon \left(z\right)+\sigma_{eig}\left(z\right)\right) & ,\;if & \sigma \left(z\right)\leq \sigma_{Fd}\left(z\right) \end{array}\right\}$$

E describes the Young’s modulus and $k_{E}$ the ratio between Young’s and *p* modulus. The *p* modulus defines the slope of the stress in the elasto-plastic area of the stress–strain diagram. $\sigma_{Fz}$ and $\sigma_{Fd}$ are the yield stresses in tension and compression. Considering the assumptions from [19], the bending moment can be determined from the stress distribution with b as wire width and d as wire thickness:(8)$$M_{B}=2\cdot b\cdot \int_{0}^{d/2}z\cdot \sigma \left(z\right)dz$$

After the wire has reached the point of maximum curvature in the straightening triangle, it is relieved again until it leaves the straightening triangle. To determine the curvature $\kappa_{2}$ remaining in the wire after the straightening process, a stress-relief cycle is calculated. For this purpose, it is assumed that the unloading stress is distributed linearly over the wire thickness, since the material does not undergo any plastic deformation during relieving. Thus, the relieving moment $M^{*}$ corresponds to the load moment:(9)$$M^{*}=M_{B}$$

From this, the unloading stress $\sigma^{*}$ can be calculated:(10)$$\sigma^{*}\left(z\right)=12\cdot \frac{M^{*}}{bd^{3}}\cdot z$$

The difference between the load stress and the unload stress in the wire after elasto-plastic loading remains as the resulting residual stress:(11)$$\sigma_{eig}\left(z\right)=\sigma \left(z\right)-\sigma^{*}(z)$$

The strain profile $\varepsilon_{2}(z)$ remaining in the wire is composed of the elastic and plastic strain components and is linear according to the assumptions made:(12)$$\varepsilon_{2}\left(z\right)=\varepsilon_{el}\left(z\right)+\varepsilon_{pl}\left(z\right)$$

The elastic strain components result from the relationship for the linear–elastic material behavior:(13)$$\varepsilon_{el}\left(z\right)=\frac{1}{E}\sigma_{eig}\left(z\right)$$

For the determination of the residual strain $\varepsilon_{2}(z)$, the plastic strain $\varepsilon_{pl}\left(z\right)$ is missing. This can be circumvented by taking a closer look at the profile of the remaining strain. In the region around the neutral fiber, only elastic strain occurs up to the point $z_{pl}$. From the assumption of linear total strain, the remaining curvature can be calculated via the following relation:(14)$$\kappa_{2}=-\frac{\varepsilon_{el}\left(z_{pl}\right)}{z_{pl}}=-\frac{\varepsilon_{2}\left(d/2\right)}{d/2}$$

The straightening force $F_{R}$ is calculated from the equilibrium conditions and the load moment $M_{B}$. Using the equilibrium relationships for successive incremental beam elements, the relationship for the straightening force on the upper roller is derived. In this paper, under the simplified assumption of only one bending triangle, this relation simplifies to:(15)$$F_{R}=\frac{M_{B}}{x_{12}}+\frac{M_{B}}{x_{23}}$$

Here, $x_{12}$ and $x_{23}$ are the distances in the x direction between the contact points of the wire at the straightening rollers 1 and 2 and 2 and 3 (cf. Figure 7). Equation (15) shows the dependence of the straightening force on the bending moment that occurs, which depends in turn on the stress distribution that occurs (Equation (8)). The stress distribution results from the material model and from the occurring strain (Equation (7)), which is caused by the position of roller 2. The position of roller 2 is always known because it Is measured in absolute terms. Thus, the measured straightening force provides information about the material properties, which are usually unknown and can only be determined by complex material testing. Thus, the precise straightening-force measurement provides important information on the material condition of the processed material.

By formulating this model for a straightening machine with three straightening rollers, the relationships between the curvature of the wire before and after the straightening machine, as well as the straightening force occurring at roller 2, can be described. This model does not include the forming process of the punch-bending machine. Therefore, the quality parameters α and θ do not occur in the model. These parameters are used for feedback control, which will not be referred to in detail in this paper. The model was implemented in Matlab R2020b by MathWorks. All parameters used are listed in Table 1.

## 3. Design Process for the Straightening Machine

The design process of the novel straightening machine was carried out according to the guidelines and methods laid out in VDI 2206 [20]. This methodology was chosen as it provides an adequate structure for the design process of mechatronic systems. To further structure the part design, calculation, and documentation steps required to create the necessary assemblies, this was expanded upon by additionally utilizing the design process by Pahl and Beitz (cf. [21]). For this approach, shortcomings in the old machine design were analyzed and characterized and concrete requirements were derived. A functional structure was created, and corresponding solutions identified and weighted by economical aspects. After selection, the technical design process was subsequently carried out. This entailed the structural design of parts using CAD, load calculations, and dimensioning as well as creating technical drawings and documentation.

### 3.1. Design Requirements

The novel straightening-machine design had to be capable of precise force measurements. Thus, the focus of the requirements phase was put on enabling the mechanical de-coupling of the measurement mechanism from the surrounding structure. As only the horizontal roller force was to be measured, the influence of lateral forces had to be minimized in the new concept. This meant mitigating points of high friction to eliminate the parasitic forces present in the old design. Re-dimensioning and/or selecting suitable new force sensors to reduce noise and improve accuracy were also required. A force resolution of at least 2 N was mandated. To minimize the influence of elastic deformations on the force measurement, the new roller-guidance and machine-frame concepts were required to possess high degrees of stiffness. In addition to this, the existing positioning concept of the rollers was to be exchanged for a method of measurement capable of delivering accurate absolute positions, thereby reducing the set-up effort required. The accuracy required for this was demanded to be at least 0.01 mm.

These central requirements were expanded upon by measures which aimed to guarantee the exchangeability of the rollers and enable the use of different roller profiles during operation. A modular design was also required if possible. To be economically viable, the existing stepper motors and their respective driveshafts were to be reused from the old design. Exchanging the existing bearing concept was optional and to be carried out if functionally necessary or advantageous.

### 3.2. Conceptional Design

The conceptual design phase was driven by the main requirement for a de-coupled force measurement. Required changes were sorted into functional groups. These consisted of geometrical changes to the old measurement concept, selection and integration of suitable sensors, bearing and drive concepts, mechanical guidance concepts, and housing integration. In the first step, the geometrical changes to the force measurement concept were implemented. To eliminate the torque caused by the roller’s off-center mounting point, the roller axis was shifted to be congruent with the sensor axis. The roller was integrated into a forked body to accommodate this (Figure 8a,b). The resulting group of components is referred to as a roller-fork hereafter.

However, this configuration still had the disadvantage of a rigid sensor and roller-fork body connection, allowing lateral forces to be transmitted into the sensor. To resolve this issue, the threaded connection between the sensor and fork body was discarded from the existing design and replaced by a flat-surfaced plunger combined with a sensor possessing a semi-spherical contact surface. In consequence, the transfer of lateral forces was restricted. Following this step, the sensor body and plunger were integrated into a common casing forming a measurement unit. This was expanded by adding two linear bearing surfaces on the unit’s sides to enable vertical movement. A linear bearing surface was also integrated at the contact surface between the plunger and casing (Figure 8c).

In the last step concerning the geometrical changes to the measurement unit, a second linear bearing surface was added to the unit casing at bearing point B (Figure 8d). This was applied to disable rotational movement of the roller-fork. Additionally, this enabled the plunger bearing point to be removed, thus also removing a source of friction.

Though it is portrayed in a hanging configuration in the attached figures, the resulting configuration of the measurement unit is modular and enables two different possible mounting positions: above or below the flat wire. Generally, measuring the roller forces of the upper rollers or lower rollers, or a combination of both would be possible. During conceptualization, these possibilities were considered and compared mechanically. Due to the mechanical layout, the mechanical loads acting on the lower rollers are of a higher magnitude than those acting on the upper rollers (cf. [22]). While placing the units as the lower roller group below the wire would have been easier to integrate from a design perspective (fixed placement, no movement), the higher acting forces would have required sensors with higher nominal forces. As the accuracy of load cells scales with the nominal force, this would have inherently reduced measurement accuracy. Placing the measurement units above the wire plane bears the additional benefit of enabling the lower rollers to be placed in a common, fixed assembly rather than on individual mounts. As a result, the decision to mount the measurement units as the upper roller group was made and followed by the design of the machine frame and guidance units.

To accommodate the measurement units, a column frame consisting of six guidance shafts mounted to a bottom and top plate was conceptualized. The measurement units were to be attached to three guidance arms that run vertically up and down on the shafts. In order to guarantee high stiffness and low friction, linear bearings were selected and integrated into the guidance arms (Figure 8, bearing point A). Three fixed-and-loose bearing configurations were integrated into the top plate to accommodate the drive spindles. These were connected to the drive shafts at the upper end and threaded into the guidance arms on the lower end. To achieve the required absolute position measurement of the roller, laser triangulation sensors were selected and integrated into the concept by placing them below the frame’s bottom mounting plate. The necessary reflective surfaces were provided by designing the roller fork bodies accordingly. Figure 9 illustrates the final machine concept.

### 3.3. Design Implementation and Description

Implementation of the final design concept was conducted using CAD tools (SolidWorks 2022). The functional groups were compiled into five-part assemblies according to their function. These assemblies are:
Machine (column) frameForce-measurement unitsPositional measurement assembly.Lower roller assemblyDrive assemblies 

The machine frame consists of the conceptualized bottom and top plates, three pairs of opposingly mounted guidance shafts, and the attached guidance units. Additionally, the frame possesses four support columns connecting the top and bottom plate. These were added to increase stiffness, bear the upper components’ weight, and transfer the process forces acting on the drive components. This reduces load on the guidance shafts and thus aids in keeping elastic deformations low during operation. The top plate accommodates the drive assemblies consisting of the three drive spindles and the associated ball groove bearings, while the bottom plate carries the lower roller assembly with the associated four rollers.

To guarantee precise movement of the guidance units, linear ball bearings were implemented. Acting as linear drive systems, these consist of the individual guiding shaft, a ball-bearing cage, and a casing that is inserted into a drilled hole in the guidance unit. Upon inserting the guidance shaft into the bearing cage, a pre-defined state of pretension is created by the fit, making the systems free of play. Each guidance unit resides on two of these bearing systems. Vertical translation of the guidance units is achieved via threads for the spindles to run in. Guidance unit, and therefore roller travel, was limited by implementing stop blocks.

As decided in the final concept, the measurement units are mounted centrally onto the guidance units. Each measurement unit consists of a forked roller carrier and a sensor casing body. The roller itself is mounted on a threaded carrier bolt, allowing the horizontal roller position to be adjusted. Vertical guidance of the roller carrier is implemented by small-scale linear ball-bearing systems, functionally identical to those used in the guidance units. The guiding shafts are fixed to the measurement unit’s casing while the roller carrier can ride vertically on them. As per the developed measurement concept, the force sensor is placed inside a casing body and connected to the roller carrier via a flat surfaced plunger. Due to the hanging configuration of the measurement units, pre-tension between the sensor and plunger can be applied via a threaded bolt, guaranteeing play-free operation. The sensor housings follow a modular design and can accept two different sizes of sensors. This enables the use of two different sensor types with either 200 N or 500 N nominal force, which translate to force resolutions of 2 N or 5 N, respectively, significantly increasing the measurement capability when compared with the precursor machine. The resulting design is shown in Figure 10a.

The positional measurement was implemented by designing a carrier assembly that accommodates three laser triangulation sensors. It is fixed to the underside of the bottom plate, which has been equipped with a matching cutout to guarantee free travel of the laser beam paths. The selected triangulation sensors are capable of achieving a positional accuracy of 0.04 mm and repeatability of up to 1 µm. The entire new straightening machine is shown in a photograph in Figure 10b.

## 4. Experimental Studies and Results

This section presents the experimental studies and their results. First, the integration of the force measurement into the straightening machine is validated. Afterwards, the presented model approach is compared with the measurement results. The flat wire used for the experimental studies was made of high-strength steel (material no. 1.4310) with a width of 3.9 mm and a thickness of 0.4 mm.

### 4.1. Decoupling between Straightening Force and Roller Position

An essential part of the new straightening-machine design is the decoupling between the force measurement and the straightening-roller movement. In conventional straightening machines, this is not possible due to the effects of high friction. The main focus in the design of the new straightening machine was on the decoupling between these two functions. In Section 3, the design actions are described in detail. At this point, proof is provided that the chosen solution fulfills its purpose. For this purpose, the straightening rollers were moved, and the force measured at the sensor was recorded. At the time of the test, there was no flat wire in the straightening machine, so the straightening rollers could be moved freely. In Figure 11, the signals of the roller position and the straightening force of one movable straightening roller over time are shown for the old and the new machine designs. With the aid of an implemented position-control system, the straightening roller was moved upwards and downwards from the zero position. The shown force signals are unfiltered and were acquired at a measuring frequency of 100 Hz. The force signal from the old machine design was influenced by the moving straightening roller due to friction effects in the guiding elements. The amplitude of the force signal was approx. ±5 N. With the new machine design, no reactions in the force measurement to the movement of the straightening roller were detectable. The signal shows only the usual noise, which has an amplitude of approx. ±0.1 N. The size of the noise amplitude corresponds to the accuracy class of the installed force sensor. For the other two straightening rollers, the same behavior was seen. Thus, it can be stated that the constructive decoupling of the straightening forces from the roller positions was successful with respect to the sensor resolution.

### 4.2. Sensitivity of the Force Measurement

In the next test, the sensitivity of the force measurement was investigated. For this purpose, a wire sample was specially prepared. Minimal elevations were applied to the wire at intervals of about 100 mm using thin adhesive tape. Figure 12 shows the profile of the wire specimen created in this way as a photo (top view) and as a schematic sketch (side view). The adhesive tape has a thickness of about 0.09 mm. The lowest strip always has a width of 50 mm. The narrow strips at position 2 and 3 are 20 mm wide.

The aim of this investigation was to analyze the sensitivity and dynamics of the force signal. For this purpose, the prepared flat wire was pulled through the straightening machine at a speed of 5 mm/s. The straightening rollers were adjusted by 0.3 mm. Thus, contact with the wire was guaranteed, but plastic deformation of the wire specimen did not yet occur. The results of this test are shown in Figure 13. There, the force at one straightening roller is shown over time. The red line indicates the wire thickness at the straightening roller at the respective time. The force signal at the straightening roller changed at the points in time when the wire profile changed. The narrow stripes at points 2 to 3 can also be clearly identified. At these points, the force signal rose immediately. The amplitude of the force increase was almost linear with the wire thickness. A slight lag in the force signal compared with the wire profile can be seen, which is to be expected due to flexibility in the bearings. The height of the force steps can also be clearly assigned to the different wire thicknesses. This shows that the force measurements are precise and reproducible even at low deflections. The force signal also shows an oscillation that extends the noise of the force sensor. It is suspected that this was caused by a minimally fluctuating feeding speed. This must be investigated in the next steps and eliminated if possible.

### 4.3. Force Measurement during Straightening Process

In the next step, the force signal at different straightening roller positions was analyzed. For this purpose, flat wire was fed through the straightening machine at a constant speed of 5 mm/s. The roller position was set from 1 mm to 3 mm in steps of 0.1 mm. As an example, three force signals for roller positions of 1.0, 1.5, and 2.0 mm are shown in Figure 14.

The force signals are unfiltered and thus represent the raw signal of the force sensors. After starting the measurement, at t=1 s the straightening roller was adjusted to the specified value and at t=10 s the feeding unit was started. It can be clearly identified in which areas the wire was stationary in the straightening machine and where it was fed. The wire feeding increased the noise amplitude in the force signal from about ±0.08 N to about ±0.3 N. The general force level between stationary and fed wire increased slightly. This was the result of a tensile force acting in the wire due to the feeding process. Further oscillation components can also be seen in the force signals. These can already be seen in the previous test (cf. Figure 13). The exact cause remains to be investigated. Presumptions suggest that the feeding unit has a slight fluctuation with respect to the feeding speed or that small variations in the thickness of the flat wire may also be responsible for this.

### 4.4. Model Validation

To validate the model, a comparison between the measured straightening force and the straightening force calculated in the model was considered. For this purpose, the force signals from the tests in Section 4.3 were evaluated and the measured straightening force was averaged over the time of feeding. This resulted in an average straightening force for each roller position. The model was also evaluated for the same roller positions and the respective straightening forces were assigned to the roller positions. The result of these investigations are shown in Figure 15. The averaged straightening forces from the measurement and the simulatively calculated straightening forces over the roller position are illustrated with the aim of proving the precise predictability of the straightening force with the help of the model. This application is a great advantage in the development of the control algorithm. The simulation results match precisely with the measurement results. Both the shape of the curve and the absolute values are very close. The maximum deviation was 1.25 N at a roller position of 2.7 mm. Apart from this outlier, the difference between model and measurement is always less than 1 N. The root mean squared error (RMSE) of all data points is 0.62 N. This shows the good predictability of the force using the model, even though the model has been simplified again compared to the complex calculations in [1,19]. The simulation time for all 21 simulation points in Figure 15 together was only 0.01 s. This shows that this model is suitable for use in an online control algorithm, because the adjustment of the straightening rollers is expected in a frequency range below 0.1 Hz. This demonstrates the applicability of the model in a feedforward or closed-loop control, as shown in Figure 6.

## 5. Discussion and Conclusions

The design of the new straightening machine opens up great potential for new and detailed investigations of the straightening process. In particular, the precise force measurement and the absolute position measurement of the straightening rollers allow a detailed examination of the forming operation in a straightening machine. This required the constructive decoupling of the force measurement from the roller movement. The measurements showed that this step was successful. The straightening roller movement has no influence on the measured force signal. The frictional influences can be reduced to a minimum. With the aid of the absolute position measurement, the straightening-roller positions are permanently known and can be adjusted to within 0.01 mm.

The precise measurement of the straightening force has also been successfully implemented. In the static case, the straightening force can be measured with a noise amplitude of just 0.08 N. In dynamic tests with continuous wire feeding, the noise increased only slightly to 0.3 N due to process conditions. Due to this, the force signal can also be used without filtering, so that no phase shift and thus no reduction in the bandwidth of a control system occurs. In order to reduce the measurement noise generated by the feeding of the material, filtering can be helpful. A resulting phase shift is acceptable, because the adjustment of the straightening rollers is expected in a frequency range below 0.1 Hz. Other data-processing algorithms can be helpful in later analysis but are not necessary yet.

The new design of the straightening machine allows changing the force sensors in order to change the wire material to be tested. It is possible to change between force sensors with nominal loads of 200 N and 500 N. However, the straightening machine is limited in its geometrical dimensions. The roller spacing is determined by the frame and is not adjustable. This means that only specific wire dimensions can be processed. It is not possible to change to significantly thicker or thinner flat wire. The development of the straightening machine also focused on flat wire. So far, its application on round wire has not been tested. It is expected that round wires with comparable cross-sectional dimensions can also be investigated. However, these investigations are still pending. It is expected that the straightening rollers will have to be changed for that purpose.

The validation of the model shows that a precise prediction of the straightening forces is possible. The RMSE between model and measurement is only 0.62 N. Due to the simplifications made, the simulation time was reduced to less than 0.01 s per simulation step. This is a great advantage for using the model in the context of an online control system. For this purpose, it is planned to design an algorithm for application-friendly parameterization. Especially in case of a coil change to another material or to other geometrical dimensions, nowadays complex investigations are necessary to parameterize the straightening models. Afterwards, a method for inverting the model must be developed so that it can be used in a control system to predict the straightening-roller positions.

Further influences on the straightening result must also be investigated. Especially, the influence of the tensile force in the wire, which is necessary for feeding the wire through the straightening machine, has to be mentioned here. Nowadays, dancer systems are used to keep the wire under tension. However, the effect of the set dancer pressure and, thus, that of the tensile force in the wire on the straightening result is ignored. The influence of the tensile force on the straightening forces and the straightening result must be investigated and the straightening process must be adapted accordingly. It is also necessary to investigate how the tensile force in the wire affects the forming operation in the straightening machine.

## Figures and Tables

**Figure 1 sensors-23-09091-f001:**
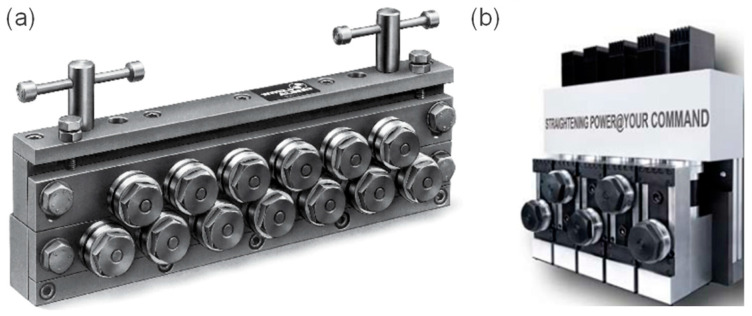
Examples of conventional wire straighteners in industrial use (**a**) [4,5] and semi-automated solutions (**b**) [6,7].

**Figure 2 sensors-23-09091-f002:**
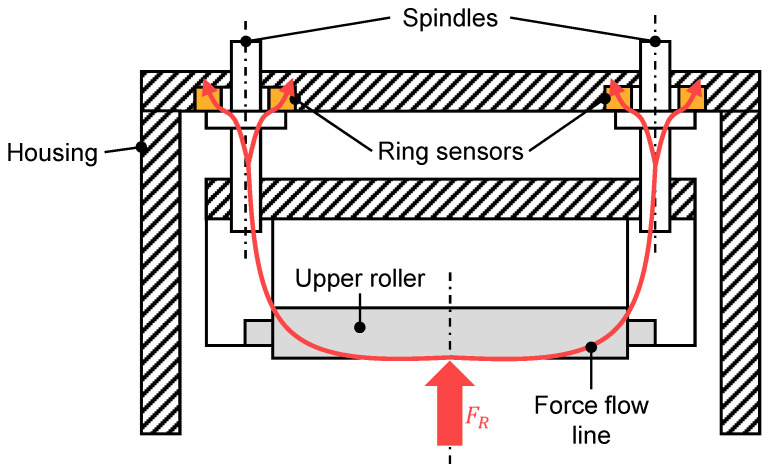
Force-measurement concept utilizing ring sensors as used by [1] (Figure based on [12]).

**Figure 3 sensors-23-09091-f003:**
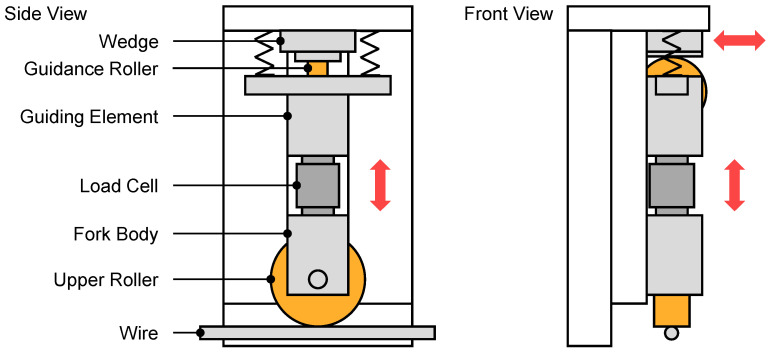
Roller-carrier concept containing load cells as used in the straightening apparatus by [14]. The lower roller rack is not illustrated.

**Figure 4 sensors-23-09091-f004:**
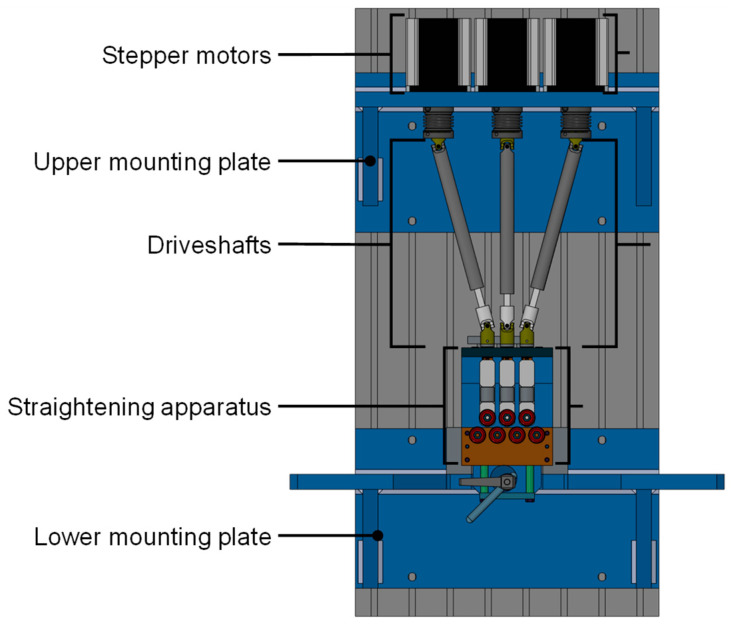
Structure of the straightening machine (CAD model) used in [11].

**Figure 5 sensors-23-09091-f005:**
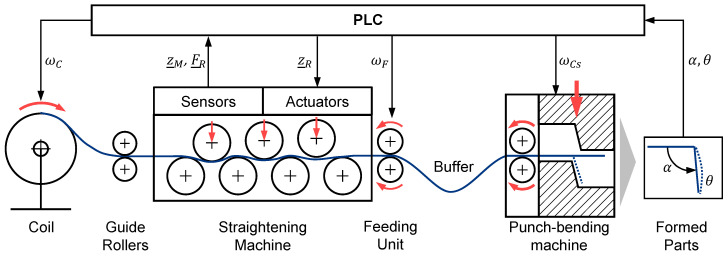
Schematic representation of the entire manufacturing process.

**Figure 6 sensors-23-09091-f006:**
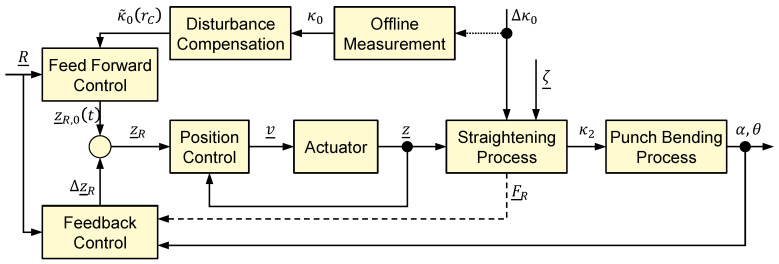
Schematic representation of the control structure.

**Figure 7 sensors-23-09091-f007:**
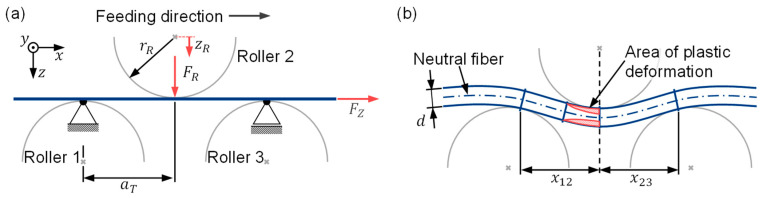
(**a**) Mechanical substitute system for one straightening triangle and (**b**) bending line of the flat wire in a straightening triangle.

**Figure 8 sensors-23-09091-f008:**
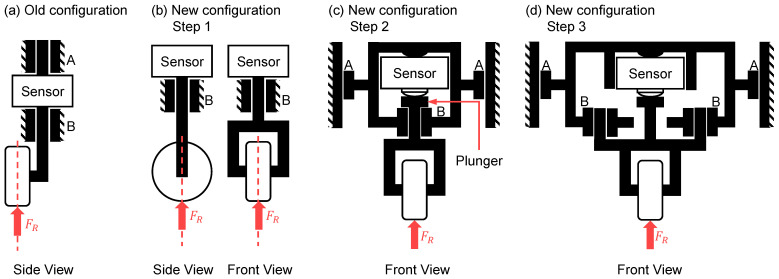
Schematic diagram of geometrical changes made to the measurement unit configuration. Bearing points are denoted as A and B. Process force is denoted as *F_R_*.

**Figure 9 sensors-23-09091-f009:**
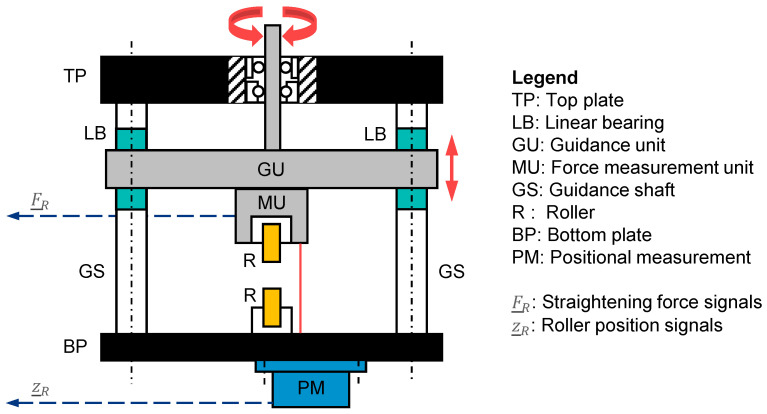
Final machine concept and its functional units after the conceptional development phase, in front view.

**Figure 10 sensors-23-09091-f010:**
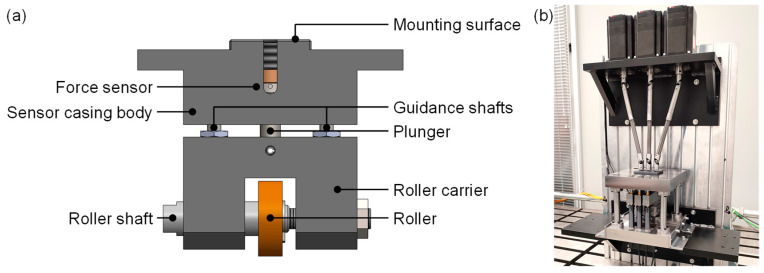
(**a**) CAD-Assembly of the developed measurement unit design implemented in the novel straightening machine; (**b**) Photo of the assembled straightening machine.

**Figure 11 sensors-23-09091-f011:**
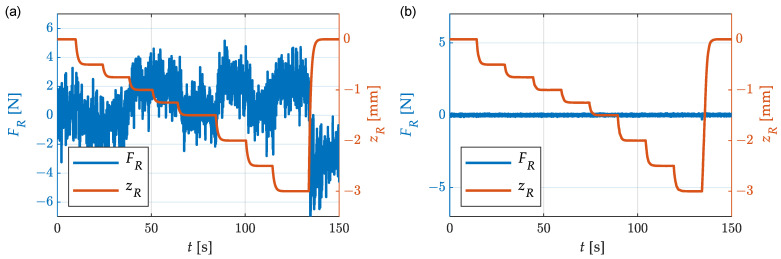
Straightening force *F_R_* during the movement of straightening roller *z_R_* without flat wire in straightening machine: (**a**) Old machine design developed in [11]; (**b**) New machine design presented in this paper.

**Figure 12 sensors-23-09091-f012:**
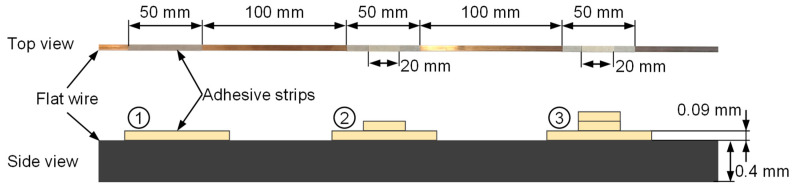
Preparation of the wire specimen with adhesive tape.

**Figure 13 sensors-23-09091-f013:**
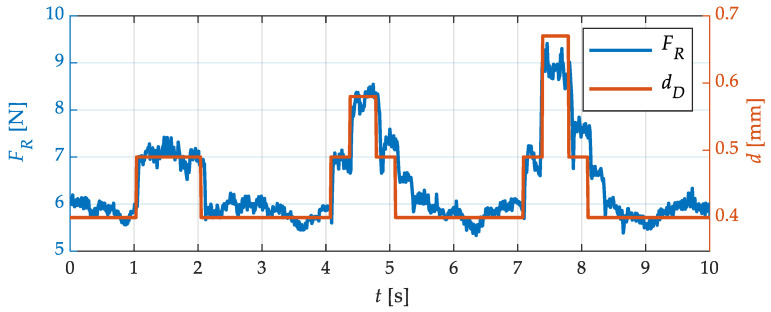
Straightening force *F_R_* on specially prepared wire specimen with changing wire thickness *d*.

**Figure 14 sensors-23-09091-f014:**
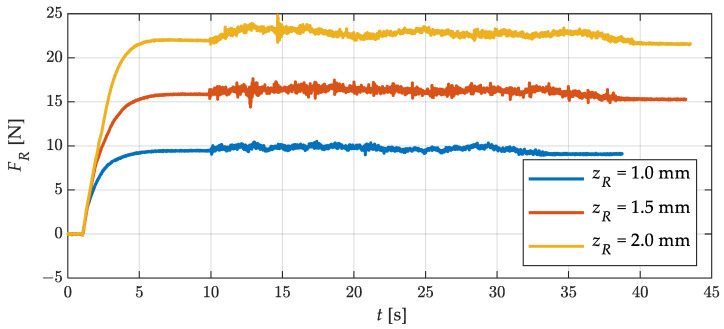
Straightening force *F_R_* at different straightening-roller positions *z_R_* during straightening process.

**Figure 15 sensors-23-09091-f015:**
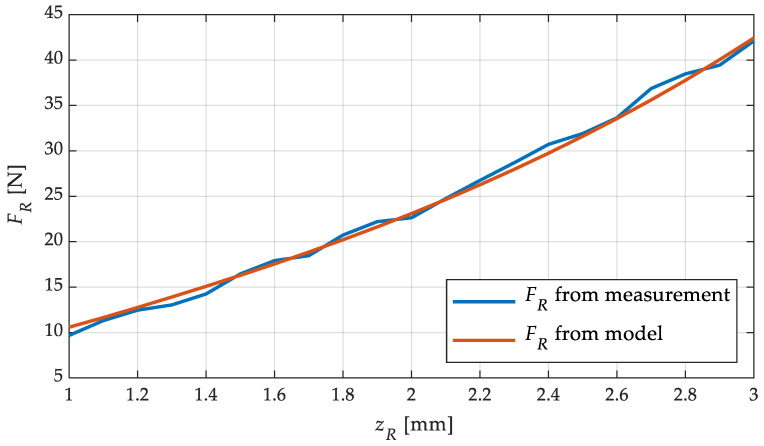
Straightening force *F_R_* from measurement and simulation for different straightening-roller positions *z_R_*.

**Table 1 sensors-23-09091-t001:** Model parameters for simulation.

Parameter	Symbol	Value	Unit
Weighting factors	$g_{1}$, $g_{2}$, $g_{3}$	1	[−]
Young’s modulus	E	95,000	MPa
*p* modulus	P	80,000	MPa
Wire width	b	3.9	mm
Wire thickness	d	0.4	mm
Yield stresses	$\sigma_{Fz}$, $\sigma_{Fd}$	±665	MPa
Straightening-roller spacing	$a_{T}$	13	mm
Radius of straightening rollers	$r_{R}$	10.8	mm

## Data Availability

Data are contained within the article.

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
