# Peer review of "Novel Straightening-Machine Design with Integrated Force Measurement for Straightening of High-Strength Flat Wire"

_sensors, 2023, doi:10.3390/s23229091_

Round 1
Reviewer 1 Report
Comments and Suggestions for Authors
1.Explain the necessity of the work undertaken, as well as the challenging issues and obstacles within the current field.
2.Provide a detailed explanation of state variables of the straightening operation mentioned in the abstract, elucidating its significance in achieving control over straightening operation and explaining how state variables can be obtained through position and how straightening operation can be controlled through state variables.
3.Can the measured data be directly used, or is it necessary to design data processing algorithms to handle the collected original data?
4.Demonstrate that the computational efficiency in the experimental results meets the requirements for online straightening operation.
5.Present a more detailed validation of the precision of the experimental results.
Comments on the Quality of English LanguageThe expression and usage of the English language are appropriate.
Reviewer 2 Report
Comments and Suggestions for Authors
The present paper proposes an approach that aims to integrate a precise force and position measurement system into a wire straightening machine. The motive for the research was well presented, and the research problem was clearly stated and described. The literature review was elegantly highlighted, thus clearly exposing the existing gap in the prior art. In my opinion, the paper is well written, and attention should be paid only to the below comments:
-
Check lines 112 to 116 for sentence repetition.
-
The authors claim that the entire straightening machine is novel, but I'm not so sure, as evidenced in refs [8 - 10]. Instead, in my opinion, the major novelty introduced is the measurement system incorporated into the present machine. By saying the entire machine is novel implies that the entire machine is based on a completely new design, which is not necessarily the case. Consequently, I will suggest that the novelty be crafted as follows: A straightening machine with a novel force and position measurement system
-
In my opinion, this paper is best suited for submission as a patent. I must confess that the paper is well written and detailed and will merit a patent.
-
Above all, my main concern is that there was no comparative analysis between the proposed system and existing mechanisms, particularly to demonstrate the advantage of using the newly introduced measurement system over existing systems. Consequently, it is difficult to appreciate any major contribution to knowledge.
-
Furthermore, it is not clear what the aim of the results in Section 4.4 is. Are the authors establishing the viability of their model? If so, I must say it is difficult to appreciate this because the experimental results were obtained from a system that was built upon the model. Therefore, similar outputs between the model and experiment should be expected.
Reviewer 3 Report
Comments and Suggestions for Authors
The authors introduced a force monitoring algorithm for motion control of a straightening process. The topic is apparently important for quality control of process industry. And their control method is interesting to give solutions to quality fluctuation. However, some concerns need to be addressed.
1) Since the adjustment is based on measured roller forces, what if the wire contains composite material/ flaw structure/.. where the stress may not be uniform?
2) Please indicate which roller acts as the monitor, and how do you set the criteria for adjusting the bending machine?
3) what is the reference position Zr? please mark the adjusting process in figure 5/7.
4) It seems the rotational speeds wc\wF\wCs can be adjustable as they controlled by PLC. While they are not clearly presented and included in Fig.6. What are the effects bringing by those speeds?
5) What are the significance of Fig.11, as it dosent reflect real process with bending wire?
6) Since you mentioned about the camera for inspecting the quality parameters (? and ?), which should be an effective way, then why not use these values for the quality analysis?
7) What are the parameters affecting the straightening process? You can add some pictures showing the influential control/ operational parameters and their imposing effects on the process.
Comments on the Quality of English LanguageClear and accurate.
Round 2
Reviewer 1 Report
Comments and Suggestions for Authors
Accept in present form
Reviewer 3 Report
Comments and Suggestions for Authors
The authors presented a good revised version, which addressed most of my concerns. I encourage authors to carry out more interesting research for further optimization of the straightening process, including quality control, process monitoring, and prediction.